# Epidemiology of Non-SARS-CoV2 Human Coronaviruses (HCoVs) in People Presenting with Influenza-like Illness (ILI) or Severe Acute Respiratory Infections (SARI) in Senegal from 2012 to 2020

**DOI:** 10.3390/v15010020

**Published:** 2022-12-21

**Authors:** Modeste Name Faye, Mamadou Aliou Barry, Mamadou Malado Jallow, Serigne Fallou Wade, Marie Pedapa Mendy, Sara Sy, Amary Fall, Davy Evrard Kiori, Ndiende Koba Ndiaye, Deborah Goudiaby, Arfang Diamanka, Mbayame Ndiaye Niang, Ndongo Dia

**Affiliations:** 1Département de Virologie, Institut Pasteur de Dakar, Dakar 12900, Senegal; 2Département de Biologie Animale, Faculté des Sciences et Techniques, Université Cheikh Anta DIOP de Dakar, Dakar 12900, Senegal; 3Epidemiology, Clinical Research and Data Science Department, Institut Pasteur de Dakar, Dakar 12900, Senegal; 4Ecole Supérieure des Sciences Agricoles et de l’Alimentation (ES2A), Université Amadou Makhtar MBOW de Dakar (UAM), Dakar 12900, Senegal

**Keywords:** respiratory infections, human coronaviruses, epidemiology, ILI, SARI, coronavirus characterization, HCoV-OC43, HCoV-NL63, HCoV-229E, HCoV-HKU1

## Abstract

In addition to emerging coronaviruses (SARS-CoV, MERS, SARS-CoV-2), there are seasonal human coronaviruses (HCoVs): HCoV-OC43, HCoV-229E, HCoV-NL63 and HCoV-HKU1. With a wide distribution around the world, HCoVs are usually associated with mild respiratory disease. In the elderly, young children and immunocompromised patients, more severe or even fatal respiratory infections may be observed. In Africa, data on seasonal HCoV are scarce. This retrospective study investigated the epidemiology and genetic diversity of seasonal HCoVs during nine consecutive years of influenza-like illness surveillance in Senegal. Nasopharyngeal swabs were collected from ILI outpatients or from SARI hospitalized patients. HCoVs were diagnosed by qRT-PCR and the positive samples were selected for molecular characterization. Among 9337 samples tested for HCoV, 406 (4.3%) were positive: 235 (57.9%) OC43, 102 (25.1%) NL63, 58 (14.3%) 229E and 17 (4.2%) HKU1. The four types circulated during the study period and a peak was noted between November and January. Children under five were the most affected. Co-infections were observed between HCoV types (1.2%) or with other viruses (76.1%). Genetically, HCoVs types showed diversity. The results highlighted that the impact of HCoVs must be taken into account in public health; monitoring them is therefore particularly necessary both in the most sensitive populations and in animals.

## 1. Introduction

Coronaviruses are highly contagious pathogens that infect a wide range of vertebrates and cause respiratory, enteric, or less frequently, neurological diseases [1,2,3], and have in recent years shown an enormous pandemic potential. In December 2019, a cluster of pneumonia cases was caused by a newly identified β-coronavirus, in Wuhan, Hubei Province in China. On the 11th of February 2020, the disease was named coronavirus disease 2019 (COVID-19 ) by the World Health Organization (WHO) and the name SARS-CoV-2 was proposed by the Coronavirus Study Group (CSG) of the International Committee [4]. As of the 11th of October 2022, more than 619 million COVID-19 cases and 6.5 million deaths have been reported globally (https://covid19.who.int/, accessed on 11 October 2022). Senegal declared its first case of COVID-19 on the 2nd of March 2020 [5], and since then the country has experienced three epidemic waves, with 1886 deaths reported since the beginning of the pandemic [6].

Coronaviruses are classified under the order Nidovirales, suborder Cornidovirineae, family Coronaviridae, subfamily Orthocoronavirinae [7]. With a single-stranded, positive-sense RNA (ssRNA) genome containing approximately 27 to 32 kilobases (kb), coronaviruses have the largest viral RNA genome described to date. Originally divided into three groups, based on their antigenic cross-reactivities and nucleotide sequences [8,9], they have recently been reclassified by the International Committee on Taxonomy of Viruses into four genera (Alphacoronavirus, Betacoronavirus, Gammacoronavirus and Deltacoronavirus) on the basis of their phylogenetic relationships and genomic structures [9]. While the alphacoronaviruses and betacoronaviruses only infect mammals, the gammacoronaviruses and deltacoronaviruses mainly infect birds, with a few also infecting mammals [10,11]. Seven human coronaviruses (HCoVs) have been identified so far: HCoV-NL63, HCoV-229E, HCoV-HKU1, HCoV-OC43, severe acute respiratory syndrome coronavirus (SARS-CoV), Middle East respiratory syndrome coronavirus (MERS-CoV) and SARS-CoV-2. HCoV-NL63, HCoV-229E, HCoV-HKU1 and HCoV-OC43 are endemic in human populations [12,13]. The genome sizes are approximately 27.5 kb for HCoV-229E and HCoV-NL63, and more than 30 kb for HCoV-OC43 and HCoV-HKU1 [7]. They most commonly infect children, generally cause mild, self-limiting upper respiratory tract infections with symptoms, such as rhinorrhoea, nasal congestion, sore throat and fever [14]. A seasonal profile with peaks of circulation during winter seasons was reported in a few studies [15,16]. Asymptomatic infections are also reported [17]. Although coronaviruses can cause severe infections in infants, young children and elderly individuals, they were considered relatively harmless to humans until the outbreak of severe acute respiratory syndrome (SARS) in 2002 and 2003 in Guangdong province, China [18,19]. The SARS-CoV epidemic resulted in more than 8000 infections with a case fatality rate of ~10% [20]. A decade later in 2012, another highly pathogenic coronavirus, Middle East respiratory syndrome coronavirus (MERS-CoV) emerged in the Kingdom of Saudi Arabia [21]. At the end of January 2020, a total of 2519 laboratory-confirmed cases of Middle East respiratory syndrome (MERS), including 866 associated deaths (case-fatality rate: 34.3%) were reported globally (https://www.emro.who.int/, accessed on 21 March 2021). However, the unprecedented and devastating impact of the global pandemic caused by SARS-CoV2 has highlighted a critical need for a geography-specific and up-to-date molecular epidemiological understanding of coronaviruses.

In sub-Saharan Africa, data on the prevalence, seasonality, clinical profiles and genetic diversity of HCoVs are lacking. In Senegal, the few data available on the prevalence of HCoVs were obtained in the context of a global study in patients suffering from influenza-like illness [22] and in children under five years with ARI [23,24]. The aim of this retrospective study is to investigate the epidemiology and genetic diversity of four common HCoVs (HCoV-NL63, HCoV-229E, HCoV-HKU1 and HCoV-OC43) during nine consecutive years (2012–2020) of influenza-like illness (ILI) surveillance in Senegal.

## 2. Materials and Methods

### 2.1. Study Design, Sample and Data Collection

Between January 2012 and December 2020, respiratory specimens and individual-level epidemiologic data were collected as part of the sentinel influenza surveillance conducted in Senegal by the National Influenza Centre (NIC) hosted at the Institut Pasteur de Dakar (IPD). Nasopharyngeal and/or oropharyngeal swabs were collected from outpatients that met the WHO case definition of influenza-like illness (ILI), or from hospitalized patients with severe acute respiratory infections (SARI), as previously described [25]. The collected samples were placed in 2 mL of universal viral transport medium (Becton Dickinson and company, Milano, Italy) and stored at 4 °C before shipment to the IPD within 72 h. At the time of sampling, a standard questionnaire was used to record both the clinical and demographic data from the enrolled patients. The specimens were processed on arrival at the laboratory for the detection, identification and characterization of viruses, including human coronaviruses (OC43, NL63, 229E and HKU1). Note that HCoV-HKU1 monitoring only started in 2018. Finally, aliquots of each sample were also stored at −80 °C for biobanking or additional analyses.

### 2.2. Nucleic Acid Extraction and Respiratory Viruses’ Detection

Total nucleic acid extraction was performed on 200 μL of each specimen using the QIAamp Viral RNA kit (QIAGEN, Valencia, CA, USA), according to the manufacturer’s instructions. The extracted RNA was eluted in 60 μL of elution buffer and stored at −80 °C until further processing. The samples were tested using a Multiplex RT-qPCR system for the detection of HCoVs (229E, NL63, OC43 and HKU1), along with other respiratory viruses (influenza A virus, influenza B virus, human respiratory syncytial virus A and B, human adenovirus, human metapneumovirus, human parainfluenza viruses 1, 2, 3 and 4, human rhinovirus A, B and C, human enteroviruses and human bocavirus), as previously described [26].

### 2.3. Human Coronaviruses’ (HCoVs) Molecular Characterization

All SARI samples positive for HCoVs were sent for sequencing. A subset of the positive samples in ILI cases for each type of HCoVs were subjected to molecular characterization, based on three criteria: positivity for HCoVs (NL63, 229E, HKU1 and OC43), the temporal distribution of the HCoV-positive samples (02 samples by type of HCoV for each month and year), and cycle threshold (Ct) value (with lower Ct-values that were correlated with higher viral loads). 

A nested RT-PCR was used to amplify the gene encoding the surface spike glycoprotein of each HCoV type. The viral RNA from the HCoVs-positive samples was extracted and reverse transcribed to cDNA, using RevertAid First Strand cDNA Synthesis Kit (Thermo Scientific, Vilnius, Lithuania). The reverse transcription was carried out in a total reaction volume of 11 μL, consisting of 1µL dNTPs (10 mM), 1 µL Rnase inhibitor (20 u/µL), 1 µL random hexamer primer (100 µM), 2 µL 5 × reaction buffer, 1 μL of RevertAid M-MuLV reverse transcriptase (200 u/μL) and 5 µL of the RNA template under the following conditions: 25 °C for 5 min, 37 °C for 1 h, 95 °C for 2 min and at a final temperature of 4 °C. Following the reverse transcription step, the cDNA product was directly used for the PCR amplification or stored at −80 °C until used.

For the PCR amplification of the targeted fragments (1096 pb, 522 pb, 871 pb and 992 pb, respectively, for HCoV-229E, HCoV-HKU1, HCoV-NL63 and HCoV-OC43), the One Taq DNA Polymerase (New England, Biolabs, Hitchin, UK) kit was used. The first PCR was carried out in a total reaction volume of 25 μL containing 13.37 μL of nuclease free water, 0.5 μL of each primer (forward and reverse primer diluted at 10 mM), 0.5 μL of dNTP, 5 μL of PCR buffer, 0.125 μL of DNA polymerase and 5 μL of the cDNA template. The reaction mixture was amplified in a thermocycler under the following conditions: an initial denaturation step of 30 s at 94 °C, followed by 35 cycles at 94 °C for 30 s, 50 °C for 1 min and 68 °C for 1 min, followed by a final step at 68 °C for 5 min. The nested PCR was performed on the resulting amplicon for each type of HCoV, using a second primer pair targeting a smaller region of the “S” gene. The amplification was carried out in a final volume of 50 μL containing 35.75 μL of nuclease free water, 10 μL of buffer, 1 μL of dNTPs, 1 μL of each primer (forward and reverse), 0.25 μL of DNA polymerase enzyme and 1 μL of the first PCR product. The amplification products were analyzed in a 1% agarose gel stained with ethidium bromide, using 1 × TAE as the electrophoresis running buffer. Following the separation of the PCR products, the amplified DNA fragments were cut and purified with the NucleoSpin^®^ Gel and PCR Clean-Up (Macherey-Nagel GmbH & Co. KG, Düren, Germany), according to the supplier’s protocol and sent to GENEWIZ (Hope end, Takeley CM22 A Essex, United Kingdom) for the bidirectional sequencing with primers used in the nested PCR. Data were then sent to the laboratory in FASTA format for analysis.

### 2.4. Phylogenetic Analysis

The sequences successfully obtained were initially cleaned when needed using the GeneStudio software (GeneStudio™ Pro, version: 2.2.0.0, accessed on 8 November 2011) and were compared to the HCoV reference strains available in GenBank using the Basic Local Alignment Search Tool (BLAST) program (www.ncbi.nlm.nih.gov/Blast.cig, accessed on 8 November 2011). Multiple alignments of the nucleotide sequences were performed using the ClustalW alignment program within the BioEdit software [27]. MEGA version 7 software [28] was used for constructing the maximum likelihood tree using the Tamura–Nei evolutionary model. The robustness of the tree topology was assessed with 1000 replicates and bootstrap values greater than 70% are shown on the consensus tree branches.

### 2.5. Statistical Analysis

The data analysis was performed using R software (R.3.0.1 version), using chi-square (χ^2^) and Fisher’s exact tests to support the comparisons of the categorical data, where *p*-values < 0.05 were considered statistically significant. The proportions were reported with 95% confidence intervals (CIs).

### 2.6. Ethical Considerations

This study was conducted within the remit of the 4S Network (Syndromic Sentinel Surveillance in Senegal), which has the approval from the Senegalese National Ethical Committee of the Ministry of Health as being less than minimal risk research. The samples and data are collected with verbal consent, as part of the clinical care and the need for written consent forms has been waived. The data is available in real-time to the Epidemiology Department at the Senegalese Ministry of Health and Prevention to support the appropriate public health action.

## 3. Results

### 3.1. Clinical and Demographic Characteristics of the Enrolled Patients

The clinical and demographic characteristics of the patients with ILI/SARI are described in Table 1. A total of 9337 patients with ILI/SARI were enrolled between January 2012 and December 2020. The highest number of enrolled patients was recorded in 2017 (2341/; 15.3%; CI:[14.7%, 15.8%]) and the lowest in 2012 (1213/; 7.9%; CI:[7.5%, 8.3%]). For the remaining years, enrolment remained relatively constant (Table 1). The male:female sex ratio was 0.99:1.0 (7620:7660). The age of the enrolled patients in this study ranged from 1 month to 95 years with a median age of 5 years.

### 3.2. Detection of HCoVs among the ILI/SARI Patients

Among the samples received during the nine-year study, 9337 were tested for the presence of HCoVs, including ILI 8773 (93.9%) CI:[93.4%, 94.4%], and SARI 564 (6.1%) CI:[5.6%, 6.5%]. HCoVs were detected in 406 patients (4.3%) CI:[3.9%, 4.8%], of which 235 (57.9%) CI:[52.9%, 62.7%] were positive for HCoV-OC43, 102 (25.1%) CI:[21%, 29.6%], 58 (14.3%) CI:[11%, 18.1%] and 17 (4.2%) CI:[2.4%, 6.6%] were positive for HCoV-NL63, HCoV-229E and HCoV-HKU1, respectively. Of the patients with HCoV infections, 369 were from ILI patients (369/8773; 4.2%) CI:[3.8%, 4.6%] and 37 from SARI patients (37/564; 6.6%) CI:[4.7%, 8.9%] (*p*-value = 0.014). The age of patients infected with HCoVs in this study ranged from 1 month to 74 years, with median ages of 3 years (interquartile range [IQR]:13.5 years). No significant difference was observed in the detections between males and females (*p*-value = 0.80). The positivity rate for HCoVs was highest in 2019 with 7.1% CI:[6.6%, 7.6%] and lowest in 2020 and 2013, with 1.8% CI:[1.5%, 2%] and 2.6% CI:[2.3%, 3%], respectively.

HCoV infections were detected in patients of all ages but with significant differences in detection rates between age groups (*p*-value *= p* < 0.005). Children under five years old were the most infected group, representing more than half of the overall HCoV positive patients (56.2%) CI:[51.2%, 61%], followed by adult patients (25–50 years old) and young children (5–10 years old), representing 13.3% (54/406) CI:[10.1%, 17%] and 10.3% (42/406) CI:[7.6%, 13.7%] of detections, respectively. The HCoVs detection rate was lowest in patients ≥ 50 years old with 1.9% (8/406) CI:[0.8%, 3.8%]. The different HCoV species were encountered in all age groups with the exception of HCoV-HKU1, which was undetected in patients between 5 and 10 years old (Table 2).

HCoV infected patients most commonly reported fever (85.5%) CI:[81.6%, 88.7%] cough (80%) CI:[75.8%, 83.8%], headache (12.3%) CI:[9.3%, 15.9%] and myalgia (10.8%) CI:[8%, 14.3%] (Figure 1 and Table 2).

### 3.3. Co-Infection of HCoVs with Other Pathogens

Samples from HCoV-positive patients were also tested for other common respiratory pathogens. Among the 406 ILI/SARI HCoV detections, 309 (76.1%) CI:[71.6%, 80.2%] were co-detected with at least one other common respiratory virus. The most frequently co-infecting pathogens were influenza viruses (159), adenovirus (118), and rhinovirus (90). Detections of multiple HCoV species were also encountered in five cases (2 SARI and 3 ILI): two cases of OC43/NL63 co-detection (1 SARI and 1 ILI), two cases of 229E/NL63 co-detection (1 SARI and 1 ILI) and one case of OC43/NL63/229E co-detection (ILI).

### 3.4. Seasonality of the HCoV Infections

The temporal distribution of the detections of the different HCoV species is illustrated in Figure 2. HCoV-OC43 (the most frequently detected) had the highest detection rates between November and January, during the six first years (2012 to 2017) of the study period, with lower activity during the remaining years (2018–2020). HCoV-NL63 was also detected throughout the study period with higher detection rates between September and January of each year. HCoV-229E has been identified sporadically in all years except in 2019, and with no distinct peaks. The HCoV-HKU1 testing began in 2018, and we noted a relatively very low detection rate without any clear seasonal pattern. 

### 3.5. Phylogenetic Analysis

During the nine-years of surveillance, fragments of the spike protein gene were successfully obtained from 53 HCoV-positive samples, including 22 sequences for HCoV-OC43, 17 for HCoV-NL63, 8 for HCoV-HKU1 and 6 sequences for HCoV-229E.

The phylogenetic analysis showed that of the 22 sequences of HCoV-OC43, 18 (81.8%) belonged to genotype G, 3 (13.6%) to genotype H and only 1 (4.5%) sequence clustered to genotype F (Figure 3A), whereas all the 6 HCoV-229E sequences obtained in this study belonged to genotype 4 (Figure 3B). With regards to HCoV-NL63, 11 (64.7%) sequences were identified as belonging to genotype A and 6 (35.3%) to genotypes B (Figure 4B). Among the eight sequences obtained for HCoV-HKU1, three (37.5%) clustered to genotype B and five (62.5%) to genotype A (Figure 4A).

## 4. Discussion

Compared to the influenza virus and respiratory syncytial virus (RSV), common HCoVs have been relatively understudied, due to their historically mild phenotype [29]. The emergence of highly pathogenic HCoVs, including SARS-CoV in 2003, MERS-CoV in 2012 and most recently, SARS-CoV2, has brought renewed interest and concern for this virus family [30]. However, to the best of our knowledge, there is very limited reporting of the molecular epidemiology of HCoVs in sub-Saharan Africa. This retrospective study is one of the first to describe the epidemiology of HCoVs in West Africa, including the phylogenetic characteristics of the HCoVs strains circulating in Senegal. During the nine years of the influenza-like illness (ILI) surveillance, a total of 9337 respiratory specimens were tested for the presence of HCoVs, with an overall prevalence of 4.3% CI:[3.9%, 4.8%]. This positivity rate is close to those reported in previous studies conducted in other countries [29,30,31,32], although lower than others reported by authors, for example, in Australia [33] and in Germany [34], with positivity rates of 8.3% and 11%, respectively. A lower detection rate of 0.9% was reported in Hong Kong in patients with acute respiratory illness over a six-year epidemiological study [35]. The variation in the HCoV detection rates among patients with ILI/SARI in different areas are likely to be explained by a range of different contextual factors, including the use of different diagnostic methods and the number of targeted species. For instance, the nested PCR and real-time PCR have be shown to be more sensitive in detecting HCoV-OC43 and 229E, when compared to the conventional detection assays [35,36]. Moreover, the differences can also validly be attributed to the geographical differences in the overall burden, differences in study populations (outpatients or hospitalized patients), the number of patients tested, the sampling period, and even the duration of the study. Consistent with several reports [35,37,38,39,40], HCoV-OC43 was the most prevalent among the four HCoV species in our study, followed by HCoV-NL63 and HCoV-229E. A seroconversion study in hospitalized children noted a similar trend [41]. Indeed, Dijkman et al. reported that HCoV-OC43 and HCoV-NL63 might induce a cross-immunity, and therefore reduce the subsequent number of clinically identified HCoV-HKU1 and HCoV-229E infections [41]. Overall, HCoV detections were more likely among ILI patients, compared to SARI patients (*p*-value = 0.03), consistent with many studies describing the association of HCoVs with mild or moderate upper respiratory tract infections [41,42,43]. However, given the syndromic approach to the case findings in our surveillance system, additional studies are required to understand the spectrum and size of the burden of HCoV infections in Senegal, including their range of severity and potential non-respiratory presentations. 

Similar to other respiratory pathogens, including RSV [44], human metapneumovirus (HMPV) [26] and adenovirus viruses [45], HCoVs were detected in all age groups, but with children under five years represented the most infected group. Consistent with our findings, several studies have reported the propensity of children under five years of age to infection with HCoVs [46,47,48,49]. However, in our data, the frequency of infections mostly reflects that the majority of patients enrolled overall with ILI or SARI, were children (more than 50% were children under 5 years old). Only 1.9% CI:[0.8%, 3.8%] of HCoV detections in our study were in patients ≥ 50 years old. In contrast, Yip et al. reported the highest detection rates of HCoVs among patients aged over 80 years old, in a descriptive epidemiological study of HCoV in hospitalized patients in Hong Kong [35]. The elderly and young children are usually considered more vulnerable to respiratory infections because of their weakened immune system and immature immunity, respectively [35]. Our study noted no influence of the patients’ gender on HCoVs infection (*p* > 0.05), despite other studies reporting higher detection rates of HCoV-OC43 and HCoV-NL63 among male patients [47].

The simultaneous screening for common respiratory viruses allowed us to investigate possible co-detections with other viral pathogens. In this study, the detection of at least one other respiratory virus occurred in 76.1% CI:[ 70.5%, 79.1%] of HCoV-positive patients, similar to those with HMPV infections in Senegal, reported previously [26]. Influenza, adenovirus and human rhinovirus were the most commonly co-detected pathogens. This is consistent with previous reports showing a high co-infection rate and similar co-infection patterns between pathogens [50,51].

Overall, our study shows that HCoVs appear to follow a seasonal pattern, with a peak of infections occurring from November through January each year coinciding with the lowest temperatures and humidity in Senegal. These findings are consistent with the results of other studies around the world [29,47,52,53]. However, unlike our results, several studies have reported a different circulation pattern of HCoVs. This is the case in Ghana where highest HCoV activity was observed during Harmattan and the rainy season (June to September) mainly for HCoV-229E and HCoV-NL63, respectively [54], and in Malaysia with detection peak between June and October [55].

The phylogenetic analysis of these Senegalese samples, based on the spike gene, illustrated the genetic diversity of the HCoV strains [55], as has been seen in other geographical areas. During the nine-year surveillance period, the co-circulation of different genotypes was noted for HCoV-OC43 (genotype G, H and F), HCoV-NL63 (genotype A and B) and HcoV-HKU1 (genotype B and A), with a predominance of genotype G for HCoV-OC43, genotype A for HCoV-NL63 and HCoV-HKU1, whereas a circulation of a single genotype (genotype 4) was observed for HCoV-229E [19,55].

However, we observed some limitations in our study. First, only a small number of HCoVs were sequenced, which creates a bias in the real relative prevalence between subtypes alongside the years of study. Furthermore, it would be interesting to carry out analyses with HCoVs ‘whole genomes for more acuity on the genetic dynamic of the circulating strains. Secondly, in our study, only few SARI patients were recruited. Thus further studies including both outpatients and more inpatients (hospitalized) would be required to firmly establish the burden associated with seasonal coronaviruses in Senegal, and clear correlations between disease severity and subtypes. 

## 5. Conclusions

This study provides an overview of the epidemiology of human coronaviruses, as well as the phylogenetic profiles of the HCoV strains circulating in Senegal. Our findings identified the four HCoV species (HCoV-OC43, HCoV-NL63, HCoV-229E and HCoV-HKU1) as common among pediatric patients with ILI/SARI, especially in children under 5 years old, and are often detected alongside other respiratory pathogens. We identified a clear circulation pattern of HCoVs, with most detections occurring between November and January. Phylogenetic analyses of the HCoV species highlighted their genetic diversity and co-circulation during the nine-year surveillance period, with the exception of HCoV-229E.

## Figures and Tables

**Figure 1 viruses-15-00020-f001:**
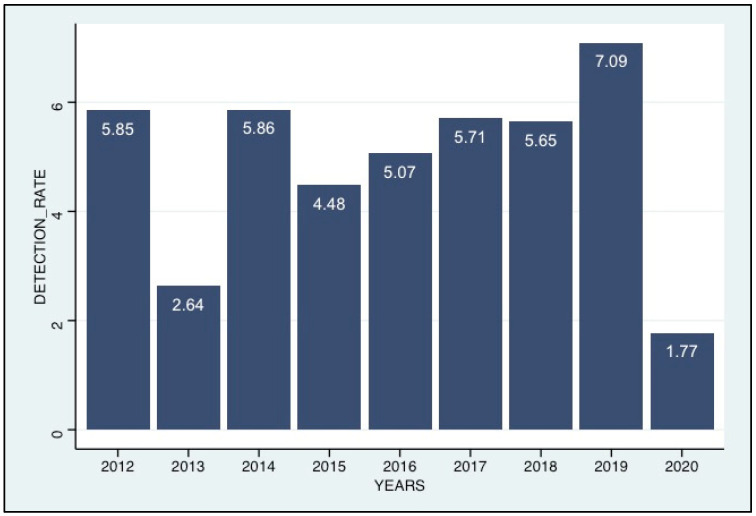
Annual trends in HcoV positivity rates (positives/samples tested), 2012–2020.

**Figure 2 viruses-15-00020-f002:**
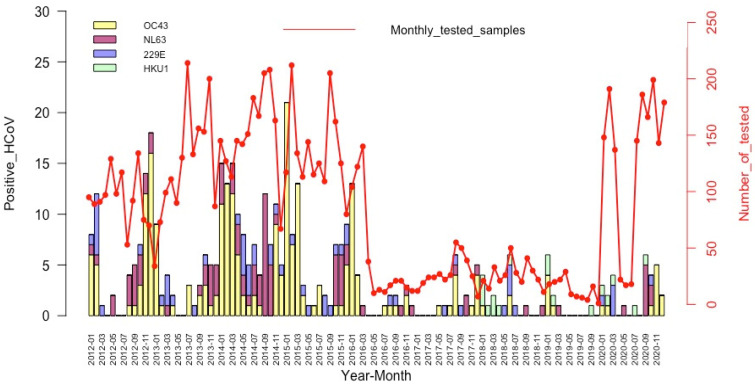
Temporal distribution of the HCoV detections among patients with an influenza-like illness, by month and year.

**Figure 3 viruses-15-00020-f003:**
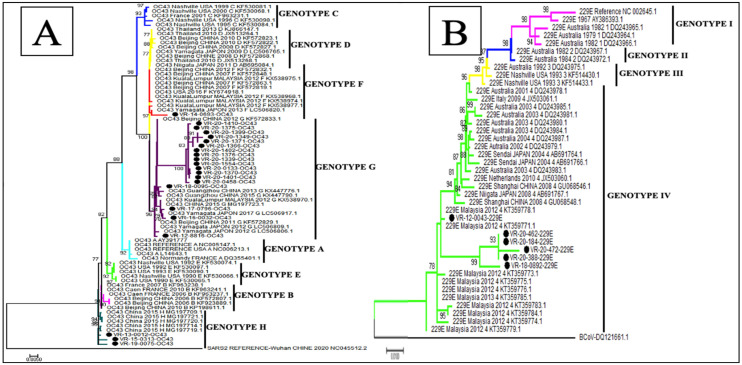
Phylogenetic analyses of HCoV-OC43 (**A**) and HCoV-229E (**B**) strains detected in patients with ILI/SARI, between 2012 to 2020, in Senegal. The phylogenetic trees are based on nucleotide sequences of the spike protein gene generated using the maximum likelihood tree with the Tamura–Nei evolutionary model, as implemented in MEGA 7 software. One thousand bootstrap replicates were performed to determine the consensus tree presented in this figure, and support for the nodes present in greater than 70% of the trees are annotated. Senegalese strains are represented in black dots.

**Figure 4 viruses-15-00020-f004:**
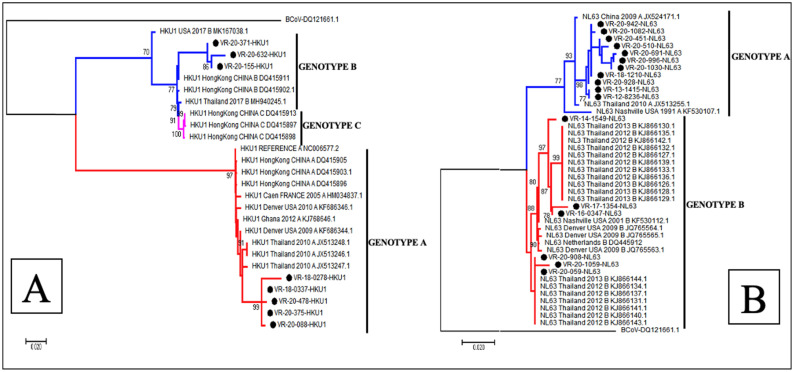
Phylogenetic analyses of HCoV-HKU1 (**A**) and HCoV-NL63 (**B**) strains detected in patients with ILI/SARI, between 2012 to 2020, in Senegal. The phylogenetic trees are based on the nucleotide sequences of the spike protein gene generated using the maximum likelihood tree with the Tamura–Nei evolutionary model, as implemented in MEGA 7 software. One thousand bootstrap replicates were performed to determine the consensus tree presented in this figure, and support for nodes present in greater than 70% of the trees are annotated. Senegalese strains are represented in black dots.

**Table 1 viruses-15-00020-t001:** Clinical and demographic characteristics of the patients with ILI/SARI in Senegal, 2012–2020.

**Year**	**2012**	**2013**	**2014**	**2015**	**2016**	**2017**	**2018**	**2019**	**2020**	**TOTAL** **(100)**
**Sample, *n* (%)**	1213 (7.9)	1518 (9.9)	1929 (12.5)	1718 (11.2)	1810 (11.8)	2341 (15.3)	1779 (11.6)	1433 (9.3)	1580 (10.3)	
**Tested for HCoV *n* (%)**	1213 (13)	1518 (16.3)	1929 (20.7)	1718 (18.4)	552 (5.9)	350 (3.7)	336 (3.6)	141 (1.5)	1580 (16.9)	9337 (100)
**Positive for HCoV *n* (%)**	71 (5.8)	40 (2.6)	113 (5.9)	77 (4.4)	28 (5.1)	20 (5.7)	19 (5.6)	10 (7.1)	28 (1.8)	406 (4.3)
**Gender, *n* (%)**										
**Female**	587 (48.4)	767 (50.5)	986 (51.1)	844 (49.1)	912 (50.4)	1182 (50.5)	886 (49.8)	708 (49.4)	788 (49.9)	7660 (50)
**Male**	610 (50.3)	743 (48.9)	936 (48.5)	873 (50.8)	895 (49.4)	1158 (49.5)	890 (50)	725 (50.6)	790 (50)	7620 (49.7)
**Age Group, *n* (%)**										
**[0–5]**	749 (61.7)	757 (49.9)	941 (48.8)	849 (49.4)	979 (54.1)	1263 (53.9)	895 (50.3)	672 (46.9)	560 (35.4)	7665 (50.0)
**[05–10]**	117 (9.6)	163 (10.7)	208 (10.8)	243 (14.1)	227 (12.5)	345 (14.7)	267 (15)	204 (14.2)	144 (9.1)	1918 (12.5)
**[10–15]**	68 (5.6)	84 (5.5)	121 (6.3)	109 (6.3)	127 (7)	153 (6.5)	107 (6)	113 (7.9)	100 (6.3)	982 (6.4)
**[15–25]**	71 (5.8)	122 (8)	229 (11.9)	155 (9)	175 (9.7)	237 (10.1)	185 (10.4)	160 (11.2)	162 (10.2)	1496 (9.8)
**[25–50]**	59 (4.9)	120 (7.9)	265 (13.7)	228 (13.2)	202 (11.2)	208 (8.9)	208 (11.7)	187 (13)	286 (18.1)	1763 (11.5)
**>50**	19 (1.6)	18 (1.2)	83 (4.3)	75 (4.4)	52 (2.9)	56 (2.4)	78 (4.4)	65 (4.5)	262 (16.6)	708 (4.6)
**Hospitalization, *n* (%)**										
**Yes**	1 (0.1)	0 (0)	2 (0.1)	29 (1.7)	29 (1.6)	94 (4)	118 (6.6)	43 (3)	124 (7.8)	440 (2.9)
**No**	1209 (99.7)	830 (54.7)	1794 (93)	277 (16.1)	975 (53.9)	2160 (92.3)	1582 (88.9)	1355 (94.6)	1232 (78)	(74.5)
**Clinical Signs, *n* (%)**										
**Cough**	824 (67.9)	1098 (72.3)	1553 (80.5)	1393 (81.1)	1469 (81.2)	1945 (83.1)	1527 (85.8)	1248 (87.1)	1434 (90.8)	(81.5)
**Headache**	105 (8.7)	182 (12)	262 (13.6)	304 (17.6)	258 (14.2)	538 (23)	411 (23.1)	414 (28.9)	416 (26.3)	2890 (18.9)
**Fever**	1129 (93.1)	1349 (88.9)	1863 (96.6)	1653 (96.2)	1761 (97.3)	2247 (96)	1648 (92.6)	1387 (96.8)	1077 (68.2)	(92.1)
**Myalgia**	124 (10.2)	326 (21.5)	0 (0)	301 (17.5)	280 (15.5)	0 (0)	188 (10.6)	199 (13.9)	0 (0)	1418 (9.3)

**Table 2 viruses-15-00020-t002:** Clinical and demographic characteristics of HCoV positive patients in Senegal, 2012–2020.

	OC43	NL63	229E	HKU1	TOTAL	*p*-Value
**Pathogens *n* (%)**	235 (57.9)	102 (25.1)	58 (14.3)	17 (4.2)	406 (100)	
**Gender, *n* (%)**						
**Male**	113 (48.1)	47 (46.2)	29 (50)	13 (76.5)	202 (49.7)	0.8047
**Female**	121 (51.5)	53 (52)	29 (50)	4 (23)	207 (51)	
**Age Group, *n* (%)**						2.2 × 10^−16^
**[0–5]**	139 (59.1)	47 (46.1)	32 (55.2)	10 (58.8)	228 (56.2)	
**[05–10]**	28 (11.9)	8 (7.8)	6 (10.3)	0 (0)	42 (10.3)	
**[10–15]**	6 (2.5)	5 (4.9)	2 (3.4)	1 (5.9)	14 (3.4)	
**[15–25]**	11 (4.7)	12 (11.8)	5 (8.6)	3 (17.6)	31 (7.6)	
**[25–50]**	30 (12.8)	14 (13.7)	8 (13.8)	2 (11.8)	54 (13.3)	
**>50**	2 (0.8)	4 (3.9)	1 (1.7)	1 (5.9)	8 (2)	
**Hospitalization, *n* (%)**						
**Yes**	10 (4.3)	8 (7.8)	8 (13.8)	8 (47.1)	34 (8.4)	0.03
**No**	133 (56.6)	82 (80.4)	35 (60.3)	6 (35.3)	256 (63)	
**Clinical Signs, *n* (%)**						2.2 × 10^−16^
**Cough**	190 (80.8)	76 (74.5)	44 (75.9)	15 (88.2)	325 (80)	
**Headache**	25 (10.6)	16 (15.7)	5 (8.6)	4 (23.5)	50 (12.3)	
**Fever**	199 (84.7)	87 (85.3)	51 (87.9)	10 (58.8)	347 (85.5)	
**Myalgia**	24 (10.2)	8 (7.8)	10 (17.2)	2 (11.8)	44 (10.8)	

## Data Availability

Not applicable.

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
