# Peer review of "Epidemiology of Non-SARS-CoV2 Human Coronaviruses (HCoVs) in People Presenting with Influenza-like Illness (ILI) or Severe Acute Respiratory Infections (SARI) in Senegal from 2012 to 2020"

_viruses, 2022, doi:10.3390/v15010020_

Round 1

Reviewer 1 Report

Overall

Thank you for inviting me to review this paper about the epidemiology and genetic diversity of seasonal coronaviruses in Senegal. This paper addresses an under-investigated area of respiratory virus epidemiology. With some additions and clarification to the manuscript, this will be a valuable addition to our understanding of human coronaviruses. More consideration into the description and reporting of epidemiological features are required to strengthen the manuscript. My specific comments and suggestions are:

Introduction

·         Please revise sentence 1, as COVID-19 caused a severe pandemic but not the deadliest in modern history

·         Would move the information about the size of HCoV genomes (line 50-52) below the sentence where you introduce these as human pathogens

·         Adding a brief overview of current understanding of prevalence, seasonality, natural history of HCoVs in other regions (there are a few studies available, some of which are cited later) would add context to both this paper and facilitate comparison with the literature in the discussion

·         Please specify the epidemiological factors you aim to investigate

Methods

·         Please add brief definition of ILI vs SARI, as this is important in terms of interpreting the symptom results

·         Were the outpatients seen because of the respiratory infection or was it opportunistic sampling?

·         Statistical analyses section should be expanded to describe how epidemiological parameters were measured/calculated

·         Please add that HKU1 test begun in 2018

Results

·         It would be interesting to see how many were SARI vs ILI in each year – or is this given by hospitalisations?

·         Lines 182-183 are confusing – the result presented appears to show that a higher proportion of the positives were from ILI patients rather than SARI patients, which seems likely to be an artifact of the samples size for both (there were more ILI patients). It would be more useful to see if the proportion of HCoV positivity was higher within tested ILI samples versus tested SARI samples (e.g. the proportion positive over the proportion tested within each group not overall)

·         Please revise the use of the word ‘rate’ throughout, which has a specific epidemiological definition. This could be changed to phrasing like ‘proportion positive’ or ‘proportion detected’ in most cases (e.g. Line 188). Where ‘detection rate’ is used for seasonality, this should be reported as ‘most commonly detected…’ or similar

·         Please report HCoV strain by year if possible

·         Report p-values to significant digits standard for the journal (usually two except where p<0.005)

·         It would be helpful to know which specific viruses were in the full panel of other viruses tested, and which strains of flu. Was COVID-19 excluded for 2020 samples?

Discussion

·         Another potential reason why detection rates may differ between studies is if different case definitions are used to decide whom to test in different studies

·         Please add strengths and limitations of the study and its design

·         The seasonality data are interesting, and some further consideration of the differences that may have led to these results differing from some other described studies would be helpful

·         The high hospitalisation rate in HKU1 may result from an outbreak

·         Consideration of how Covid-related measures may have impacted 2020 results would be helpful

·         The conclusions in the abstract (final sentence) are not given in the conclusions section in the discussion and should be expanded upon

Author Response

Reviewer 1 :

Overall

Thank you for inviting me to review this paper about the epidemiology and genetic diversity of seasonal coronaviruses in Senegal. This paper addresses an under-investigated area of respiratory virus epidemiology. With some additions and clarification to the manuscript, this will be a valuable addition to our understanding of human coronaviruses. More consideration into the description and reporting of epidemiological features are required to strengthen the manuscript. My specific comments and suggestions are:

Author’s response (AR): Thank you for the great interest you showed for this work, and the positive evaluation. I also appreciate your useful suggestions and comments which will undoubtedly help to improve the quality of this paper.

Introduction

  • Please revise sentence 1, as COVID-19 caused a severe pandemic but not the deadliest in modern history.

The sentence was rephrased removing the superlative «the deadliest».

  • Would move the information about the size of HCoV genomes (line 50-52) below the sentence where you introduce these as human pathogens.

Corrected as suggested by the reviewer.

  • Adding a brief overview of current understanding of prevalence, seasonality, natural history of HCoVs in other regions (there are a few studies available, some of which are cited later) would add context to both this paper and facilitate comparison with the literature in the discussion.

As suggested by the reviewer the HCoVs seasonality is reported in the introduction. However, we avoided end up with an overly long introduction dealing with the history of the viruses as well as prevalences reported elsewhere, especially since in the discussion this last point is largely developed.

  • Please specify the epidemiological factors you aim to investigate

As stated in the introduction we investigated the epidemiology and the genetic diversity of the seasonal coronaviruses in Senegal. And by epidemiology we meant prevalence, age group repartition, seasonality, clinical profile (severity).

Methods

  • Please add brief definition of ILI vs SARI, as this is important in terms of interpreting the symptom results

ILI and SARI are terms already defined by WHO or CDC, and these case definition are worldwide use in the global surveillance of FLU. So references are proposed for more details.

  • Were the outpatients seen because of the respiratory infection or was it opportunistic sampling?

Indeed, patients were seen because of ILI or SARI. Physicians on sites are trained to identify patients for inclusion based on clear cases definition for both ILI and SARI.

  • Statistical analyses section should be expanded to describe how epidemiological parameters were measured/calculated

More details are given in the resubmitted version.

  • Please add that HKU1 test begun in 2018

Already well stated « Note that HCoV-HKU1 monitoring only started in 2018 ».

Results

  • It would be interesting to see how many were SARI vs ILI in each year – or is this given by hospitalisations?

Indeed, SARI mean hospitalized patients.

  • Lines 182-183 are confusing – the result presented appears to show that a higher proportion of the positives were from ILI patients rather than SARI patients, which seems likely to be an artifact of the samples size for both (there were more ILI patients). It would be more useful to see if the proportion of HCoV positivity was higher within tested ILI samples versus tested SARI samples (e.g. the proportion positive over the proportion tested within each group not overall)

Indeed, the reviewer is right, it has some confusions and even mistakes in this paragraph. In this version, this part is rephrased with the right results and precisions.

  • Please revise the use of the word ‘rate’ throughout, which has a specific epidemiological definition. This could be changed to phrasing like ‘proportion positive’ or ‘proportion detected’ in most cases (e.g. Line 188). Where ‘detection rate’ is used for seasonality, this should be reported as ‘most commonly detected…’ or similar

Thank you for this useful suggestion which is taken into account in this version.

  • Please report HCoV strain by year if possible

The reviewer suggestion is taken into account in the Table 1.

  • Report p-values to significant digits standard for the journal (usually two except where p<0.005)

Done as suggested.

  • It would be helpful to know which specific viruses were in the full panel of other viruses tested, and which strains of flu.

Done as suggested by reviewer.

Was COVID-19 excluded for 2020 samples?

Indeed, we voluntary avoid to talk about COVID-19 in this paper. However, a study on the incidence of SARS-CoV-2 emergence in the epidemiology of other seasonal respiratory viruses in ongoing.

Discussion

  • Another potential reason why detection rates may differ between studies is if different case definitions are used to decide whom to test in different studies

Indeed, the reviewer is right, cases definition difference could impact detection rates in studies.

  • Please add strengths and limitations of the study and its design

Indeed, the reviewer is right, and limitations are pointed out in the resubmitted version.

  • The seasonality data are interesting, and some further consideration of the differences that may have led to these results differing from some other described studies would be helpful
  • The high hospitalisation rate in HKU1 may result from an outbreak.

It was not an oubreak, just cases detected at different moments and different sites.

  • Consideration of how Covid-related measures may have impacted 2020 results would be helpful

Indeed, it is a very interesting aspect. In fact, we are currently doing analyses in this way, in order to see the incidence of the SARS-CoV-2 emergence on the circulation profile of other respiratory viruses, and these data will be the subject of another paper. Otherwise on this work we wanted to remain focused only on seasonal coronaviruses.

Reviewer 2 Report

The paper provides an analysis of the epidemiology of four human coronaviruses in Senegal over 9 years. It is an interesting dataset that has been collected, but the analysis has limited the validity of any results. I have a few major revisions that I think are required to the paper before publication.

1) No confidence intervals have been calculated throughout the paper - the percentage of HCoV in ILI/SARI patients was reported at 4.3% why havent the 95% confidence intervals been given? They are [3.9%, 4.8%]. This should be given for all such calculations.

2) In section 3.2 the p-values have been given for lots of comparisons have been given, but the actual estimates of values (and confidence intervals) have not been given. For example HCoV positivity rate for ILI and SARI was not given, only that there was a p-value of 0.03. For age groups HCoV positivity rate was also not given (only compared raw numbers between age groups)

3) In section 3.4 the trends in positivity are discussed, but these are very hard to see in figure 2. Stacking the different HCoVs makes it difficult to see trends in a single one - consider a multipanel plot as well. Additionally, given so much focus is on the trends in a single year (months with high % etc) another figure showing the samples collected by month would be useful - maybe for each HCoV a graph showing the number detected each month (with stacked colours for year of detection). Some statistical analysis is also needed in this section to make clear what observations are statistically significant.

Also a minor point:

1) The introduction is too heavily focused on SARS, MERS and SARS-CoV-2 for a paper on HCoVs

Author Response

Reviewer 2

The paper provides an analysis of the epidemiology of four human coronaviruses in Senegal over 9 years. It is an interesting dataset that has been collected, but the analysis has limited the validity of any results. I have a few major revisions that I think are required to the paper before publication.

Author’s response (AR): Thank you for the great interest you showed for this work, and the positive evaluation. I also appreciate your useful suggestions and comments which will undoubtedly help to improve the quality of this paper.

1) No confidence intervals have been calculated throughout the paper - the percentage of HCoV in ILI/SARI patients was reported at 4.3% why havent the 95% confidence intervals been given? They are [3.9%, 4.8%]. This should be given for all such calculations.

The reviewer’s suggestions are largely taken into account in the resubmitted version.

2) In section 3.2 the p-values have been given for lots of comparisons have been given, but the actual estimates of values (and confidence intervals) have not been given. For example HCoV positivity rate for ILI and SARI was not given, only that there was a p-value of 0.03. For age groups HCoV positivity rate was also not given (only compared raw numbers between age groups)

The reviewer’s suggestions are included in the resubmitted version.

3) In section 3.4 the trends in positivity are discussed, but these are very hard to see in figure 2. Stacking the different HCoVs makes it difficult to see trends in a single one - consider a multipanel plot as well. Additionally, given so much focus is on the trends in a single year (months with high % etc) another figure showing the samples collected by month would be useful - maybe for each HCoV a graph showing the number detected each month (with stacked colours for year of detection). Some statistical analysis is also needed in this section to make clear what observations are statistically significant.

Thank you for the reviewer suggestions, but we avoided to have to many graphs in this paper, and also we want people to have a comparative figure with viruses represented in different colors in order to see the ciruculation profile of each virus during the study period.

Also a minor point:

1) The introduction is too heavily focused on SARS, MERS and SARS-CoV-2 for a paper on HCoVs

Indeed, the introduction may seem too exhastive on the viruses listed by the reviewer, but we thought it was important to highlight the real potential of these coronaviruses, and that it is important to have a focus on and better monitor these seasonal coronaviruses that infect us very regularly and, following a genetic recombination event or particular mutations,  could become a more serious public health threat.